# Chitosan–Sodium Caseinate Composite Edible Film Incorporated with Probiotic *Limosilactobacillus fermentum*: Physical Properties, Viability, and Antibacterial Properties

**DOI:** 10.3390/foods11223583

**Published:** 2022-11-10

**Authors:** Seat Ni Wai, Yu Hsuan How, Lejaniya Abdul Kalam Saleena, Pascal Degraeve, Nadia Oulahal, Liew Phing Pui

**Affiliations:** 1Department of Food Science and Nutrition, Faculty of Applied Sciences, UCSI University, Kuala Lumpur 56000, Malaysia; 2BioDyMIA Research Unit, Université Claude Bernard Lyon 1, ISARA Lyon, 155 Rue Henri de Boissieu, F-01 000 Bourg en Bresse, France

**Keywords:** antibacterial activity, bioactive, biodegradable, film properties, food packaging, polymer

## Abstract

Single-use synthetic plastics that are used as food packaging is one of the major contributors to environmental pollution. Hence, this study aimed to develop a biodegradable edible film incorporated with *Limosilactobacillus fermentum*. Investigation of the physical and mechanical properties of chitosan (CS), sodium caseinate (NaCas), and chitosan/sodium caseinate (CS/NaCas) composite films allowed us to determine that CS/NaCas composite films displayed higher opacity (7.40 A/mm), lower water solubility (27.6%), and higher Young’s modulus (0.27 MPa) compared with pure CS and NaCas films. Therefore, *Lb. fermentum* bacteria were only incorporated in CS/NaCas composite films. Comparison of the physical and mechanical properties of CS/NaCas composite films incorporated with bacteria with those of control CS/NaCas composite films allowed us to observe that they were not affected by the addition of probiotics, except for the flexibility of films, which was improved. The *Lb. fermentum* incorporated composite films had a 0.11 mm thickness, 17.9% moisture content, 30.8% water solubility, 8.69 A/mm opacity, 25 MPa tensile strength, and 88.80% elongation at break. The viability of *Lb. fermentum* after drying the films and the antibacterial properties of films against *Escherichia coli* O157:H7 and *Staphylococcus aureus* ATCC 29213 were also evaluated after the addition of *Lb. fermentum* in the composite films. Dried *Lb. fermentum* composite films with 6.65 log_10_ CFU/g showed an inhibitory effect against *E. coli* and *S. aureus* (0.67 mm and 0.80 mm inhibition zone diameters, respectively). This shows that the *Lb.*-*fermentum*-incorporated CS/NaCas composite film is a potential bioactive packaging material for perishable food product preservation.

## 1. Introduction

Among various packaging materials, plastics have gained considerable attention in the food industry with their outstanding properties that include lightweight, resilient, usually non-reactive, waterproof, and low cost. However, synthetic plastic films are usually non-biodegradable, which led to the rapidly increasing production of disposable plastic packaging that has overwhelmed the world’s ability to deal with them [1]. This environmental issue could be overcome by replacing plastic packaging with active packaging that is highly renewable and degradable [2]. The release of active compounds into packaged food could prevent the deterioration of foods during storage and distribution [3].

According to Market Research Futures (MRFR), the edible packaging market (based on protein, lipids, polysaccharides, and others) will be worth USD 2.14 billion by 2030, with a compound annual growth rate (CAGR) of 6.79 percent (2022–2030), up from USD 783.32 million in 2021. Chitosan (CS) and sodium caseinate (NaCas) received considerable attention from researchers regarding formulating edible films owing to their distinct physical and mechanical properties [4,5]. Chitosan is the second most widespread natural polysaccharide found after cellulose [6]. It is usually derived from chitin and characterized as colorless, non-toxic, biodegradable, and has antibacterial properties [7]. On the other hand, NaCas was also found to produce films with high transparency and flexibility [8]. In addition to this, NaCas films are good at regulating the migration of non-polar substances that include oxygen and carbon dioxide gas, as well as some flavors and aroma compounds [9].

A composite edible film is a combination of two hydrocolloids produced through a coating or emulsion technique [10]. Despite its many advantages, the hydrophilic character and weak mechanical stability of CS restrict its edible film application [11]. Moreover, NaCas was also reported to be a poor moisture barrier. Interestingly, sodium caseinate was found to possess excellent film-forming capabilities coupled with high mechanical strength over other biopolymers [8]. Hence, a composite edible film developed with NaCas and CS could improve the moisture barrier and mechanical stability in comparison to the single NaCas and CS edible films [12,13]. Volpe et al. [14] reported higher tensile strength and elongation at break for sodium caseinate and chitosan blended films in comparison to chitosan films. Furthermore, the polyelectrolyte complexation developed between sodium caseinate and chitosan chains lowers the moisture content and improves the strength of the blended films [15].

Various natural antimicrobial agents, such as organic acids, essential oils, and plant extracts, were incorporated into edible films [4]. The selectively permeable barrier of an edible film can maintain the maximum antibacterial effect on the packaged food surface, thereby extending the product’s shelf life. Aside from these antimicrobial agents, probiotics have also received considerable attention as a functional additive for active food packaging [16,17,18]. Among its many beneficial effects, one of the key biological functions of a probiotic is antimicrobial activity. Probiotics were reported to exert antimicrobial effects on pathogens through different mechanisms, such as competing for nutrients and bacteriocins production [19].

In comparison to plastic food packaging, probiotic film packaging gained great popularity as an emerging technology, as it could also provide additional nutritional value to food products and exert beneficial effects upon consumption [20]. *Bifidobacterium animalis* subsp. *lactis* BB12 was incorporated into whey-protein- and sodium-alginate-based edible films. The viable cell count of *B. lactis* was able to maintain above 6 log_10_ CFU/g of film for 60 days at room temperature [21]. Furthermore, *Lactobacillus plantarum* strains added in Konjac-based edible film showed anti-fungal properties and preserved fresh-cut kiwis for 5 days under refrigerated temperature [22]. Moreover, the encapsulation of probiotic cultures in edible film allows it to be stable at room temperature for more than 24 days [23].

The effectiveness of probiotic edible films for the shelf life extension of different food matrices, such as fruits, vegetables, meat, fish, and bakery, was also reported in several studies [17,24,25,26]. López de Lacey et al. [27] showed that *Lactobacillus acidophilus* and *Bifidobacterium bifidum* in an edible film was able to migrate onto fish fillets upon application and inhibited the growth of pathogens (*Shewanella putrefaciens* and *Photobacterium phosphoreum*). However, the application of edible films to food products remains a challenge, as the properties of the film may vary with the probiotic strains and biopolymer matrices being used. Hence, it is crucial to explore the impact on the edible film’s physical and mechanical properties after the incorporation of probiotics [28]. Moreover, the question of the stability of the viability of bacteria following their addition to films is also crucial for their potential applications [29]. Interestingly, Léonard et al. [29] observed that the viability of two strains of bioprotective lactic acid bacteria, including a *Lactobacillus paracasei* strain, was better maintained in composite alginate–caseinate matrices than in alginate matrices for 12 days storage at 30 °C.

*Limosilactobacillus fermentum* is one of the *Lactobacillus* species recognized as part of the human microbiota that colonizes various parts of the human body, such as the gastrointestinal tract, vagina, and mouth [30]. Several studies showed the potential of *Lb. fermentum* for use as probiotics and in the medical field because of their health-promoting effects, such as preventing alcoholic disease and reducing gastrointestinal and upper respiratory tract infections. Its anti-proliferative, immunomodulatory, anti-inflammatory, and antioxidant activities were also reported [31,32,33]. Interestingly, the antimicrobial activity of *Lb. fermentum* against *Helicobacter pylori*, *Clostridium perfrigens*, *Streptococcus mutans*, *Pseudomonas aeruginosa*, *Micrococcus luteus*, and common food pathogen *Escherichia coli* was also reported [34,35,36,37]. Its use for food preservation applications is promising, not only owing to its lactic acid production but also due to its bacteriocin production [38].

Although several publications reported on CS and NaCas films, there is limited research focused on the properties of CS/NaCas composite films and the incorporation of *Lb. fermentum* as a bioactive ingredient in such films has never been reported. Therefore, CS, NaCas, and CS/NaCas composite films were first prepared and their physical and mechanical properties were determined. Further tests on the viability and antibacterial activity of *Lb. fermentum*-incorporated films were carried out. 

## 2. Materials and Methods

### 2.1. Preparation of the Culture 

*Lb. fermentum* cells were isolated from NuvaPro, Chr. Hansen, Horsholm, Denmark and prepared according to Lee et al. [39]. One gram of *Lb. fermentum* powder was dissolved into 100 mL of sterile de Man, Rogosa, and Sharpe (MRS) broth (Merck, Darmstadt, Germany) and incubated at 37 ℃ for 24 h. The *Lb. fermentum* culture was subjected to centrifugation (AllegeraTM X-22R Centrifuge, Beckman Coulter Life Sciences, Indianapolis, IN, USA) at 1738× *g* for 10 min. The supernatant was discarded and the harvested probiotic cells were washed twice with 0.9% (*w*/*v*) NaCl solution to obtain a final cell count of 10^8^–10^9^ CFU/mL prior to usage.

### 2.2. Preparation of the Bioactive Film 

Chitosan (CS) suspension (2% *w*/*v*) was prepared by dissolving chitosan powder (>99.0%, Chemsoln, Shah Alam, Malaysia) in 1.0% acetic acid solution and stirred overnight at room temperature with a magnetic stirrer to achieve a complete dispersion of chitosan. The pH of the CS suspension was adjusted to 5.0 using 1 M NaOH and constant stirring for 15 min. Subsequently, 4% (*w*/*v*) NaCas powder (R&M Chemicals, Chandigarh, India) was added to the distilled water with constant stirring at room temperature for 4 h. Both CS and NaCas suspensions were added with 1% (*v*/*v*) glycerol and stirred for 15 min. 

The preparation of the chitosan/sodium caseinate (CS/NaCas) composite films was achieved according to Volpe et al. [14] with slight modification. The chitosan film-forming suspension (100 mL) was mixed with 100 mL of sodium caseinate film-forming suspension at a 1:1 ratio. The composite suspensions were subjected to constant stirring at room temperature by using a magnetic stirrer for 1 h and left on the bench for 30 min to eliminate the gas bubbles.

An *Lb. fermentum* suspension (1.0% (*v*/*v*)) was added to the optimized film-forming suspension (8.96 log_10_ CFU/mL). The film-forming suspension (25 mL) was cast on a sterile Petri dish and dried at 40 °C for 48 h in a ventilated oven. The dried individual or composite films were stored in desiccators for 24 h and peeled off for further analysis. 

### 2.3. Film Analysis

#### 2.3.1. Thickness

The thickness (mm) of each film sample was determined using a micrometer screw gauge. Measurements were done at seven different segments of the films to obtain the average values [18].

#### 2.3.2. Moisture Content

The moisture content (MC) of the film was evaluated according to Kuan et al. [40] with modifications. The film samples were stored in a desiccator with silica gel at 25 °C. Each film sample was cut into 2 cm × 2 cm pieces and placed in a hot air oven for 24 h at 60 °C (until a constant weight was achieved). After drying, the moisture content was calculated from the initial and final weights of the film. The moisture content was expressed as a percentage and calculated using Equation (1):(1)$$\mathrm{Moisture}\;\mathrm{content}\;\left(\%\right)=\frac{\mathrm{Weight}\;\mathrm{before}\;\mathrm{drying}\;\left(g\right)-\mathrm{Weight}\;\mathrm{after}\;\mathrm{drying}\;\left(g\right)}{\mathrm{Weight}\;\mathrm{before}\;\mathrm{drying}\;\left(g\right)}\times 100\%$$

#### 2.3.3. Water Solubility

Water solubility of the film samples was determined according to Siah et al. [41] with some modifications. Film samples (2 cm × 2 cm) were dried in a hot air oven at 60 °C for 24 h. The initial dry mass of the film was recorded. The dried film samples were immersed in 80 mL of deionized water and constantly stirred for 30 min at 25 °C. After immersion, the remaining film was taken out and blotted dry with filter paper before being dried in the oven at 60 °C for 24 h. Once a constant weight was obtained, the final weight was measured and the weight difference was considered the soluble solids. The water solubility of the film was expressed as a percentage by dividing soluble solids by the initial dry weight. The water solubility of the film was calculated using Equation (2):(2)$$\mathrm{Water}\;\mathrm{solubility}\;\left(\%\right)=\frac{\mathrm{Initial}\;\mathrm{dry}\;\mathrm{weight}\;\left(g\right)-\mathrm{Final}\;\mathrm{dry}\;\mathrm{weight}\left(g\right)}{\mathrm{Initial}\;\mathrm{dry}\;\mathrm{weight}\;\left(g\right)}\times 100\%$$

#### 2.3.4. Opacity

Film opacity was determined based on the method described by Choong et al. [42] with a minor modification. First, the film samples were cut into rectangular shapes (1 cm width and 3.5 cm long) to fit into the plastic cuvettes. The absorbance was measured at 600 nm using a UV spectrophotometer and an empty cuvette was used as a blank. The opacity of the films was calculated using Equation (3):(3)$$\mathrm{Opacity}\;=\frac{A_{600}}{\mathrm{Thickness}\;\left(\mathrm{mm}\right)}$$

#### 2.3.5. Color

The film’s color was determined using a Lab colorimeter (ColorFlex EZ, HunterLab, Reston, VA, USA) [43]. A 64 mm diameter circular sample was cut and placed in a ColorFlex sample cup after colorimeter calibration with white and black plates. The film color was expressed as L* (lightness–darkness), a* (red–green), and b* (yellow–blue) values. The total color differences between each sample ΔE were calculated based on Equation (4):(4)$$\Delta E=\sqrt{{\left(L-L^{*}\right)}^{2}+{\left(a-a^{*}\right)}^{2}+{\left(b-b^{*}\right)}^{2}}$$

#### 2.3.6. Mechanical Properties

The tensile strength (TS), the elongation at break (EAB), and Young’s modulus were determined according to the ASTM standard method 828-88, as described by Lee et al. [39] with a slight modification. First, the samples were cut into 6 cm × 1 cm strips and placed in the desiccator containing saturated silica gel at 25 °C for 48 h before testing. A texture analyzer (TA-XT2, Stable Micro Systems, Godalming, UK) was used to evaluate the TS and EAB in tensile mode with a 40 mm initial grip separation and a crosshead speed of 20 mm/min. The film was mounted on the grips and stretched until it broke. The peak load (N) and peak extension (mm) readings were obtained, while the TS, the EAB, and Young’s modulus were calculated by using Equations (5)–(7), respectively:(5)$$\mathrm{Tensile}\;\mathrm{strength}\;\left(\mathrm{MPa}\right)=\frac{\mathrm{Peak}\;\mathrm{load}\;\left(N\right)}{\mathrm{Cross}\;\mathrm{sectional}\;\mathrm{area}\;\left({\mathrm{mm}}^{2}\right)}$$
(6)$$\mathrm{Elongation}\;\mathrm{at}\;\mathrm{break}\;\left(\%\right)=\frac{\mathrm{Final}\;\mathrm{length}\;\mathrm{of}\;\mathrm{film}\;\mathrm{ruptured}\;\left(\mathrm{mm}\right)}{\;\mathrm{Initial}\;\mathrm{grip}\;\mathrm{length}\;\left(\mathrm{mm}\right)}\times 100\%$$
(7)$$\mathrm{Young}’s\;\mathrm{modulus}\;\left(\mathrm{MPa}\right)=\frac{\mathrm{Tensile}\;\mathrm{strength}\;\left(\mathrm{MPa}\right)}{\;\mathrm{Elongation}\;\mathrm{at}\;\mathrm{break}\;\left(\%\right)}$$

#### 2.3.7. Viability of *Lb. fermentum*

The viability of *Lb. fermentum* in the edible film was determined before and after the film-drying process by referring to the procedure of López de Lacey et al. [27] with a minor modification. Briefly, 1 g of the probiotic-incorporated edible films was blended with 9 mL of sterile 2% (*w*/*v*) trisodium citrate and vortexed for 30 s. The solution was mixed gently using constant agitation in a shaker incubator at 37 °C for 1 h to release the bacteria. Both before film-forming and after releasing from the edible film, the *Lb. fermentum* was subjected to serial dilution by mixing with 9 mL of 0.9% (*w*/*v*) NaCl solution. Each dilution was plated on MRS agar and incubated for 48 h at 37 °C. The enumeration of bacteria was carried out using the pour plate method [44]. The total number of viable probiotic bacteria was expressed as log colony-forming units per gram (log_10_ CFU/g).

#### 2.3.8. Antibacterial Activity 

The antibacterial activity of the *Lb. fermentum* incorporated edible films was determined using the disc diffusion method according to Abdollahzadeh et al. [4] with a few modifications. One colony of *E. coli* O157:H7 and *S. aureus* 29213 were inoculated into 5 mL of nutrient broth and incubated at 37 °C for 18–24 h to achieve approximately 10^5^–10^6^ CFU/mL. Bacterial suspensions (100 μL) were spread on Mueller Hinton (MH) agar plates. One gram of *Lb. fermentum* edible film (6.65 log_10_ CFU/g) was then dissolved in 9 mL of sterile 2% (*w*/*v*) trisodium citrate and vortexed for 30 s. The film solution was constantly stirred using a magnetic stirrer at 37 °C for 1 h to release the bacteria. A sterile disc was dipped into the solution and dried in the laminar flow. Edible films without *Lb. fermentum* were used as negative controls, antibiotic discs (tetracycline) were used as positive controls, and sterile discs were used as blank controls. The discs were placed on the MH agar spread with pathogens. The plates were then incubated at 37 °C for 24 h. The zone of inhibition formed around the films was measured.

### 2.4. Statistical Analysis

All the analyses were conducted in triplicate and expressed as mean ± standard deviation. SPSS version 25.0 software (IBM Corp., Armonk, NY, USA) was used for the statistical comparisons. One-way analysis of variance (ANOVA) with Tukey’s post hoc test was used to compare the CS, NaCas, and CS/NaCas edible films with a significance level of *p* < 0.05. On the other hand, an independent *t*-test was used to compare the edible film with or without *Lb. fermentum*, while a paired *t*-test was used to compare the total viable cell count of *Lb. fermentum* before and after the film-drying process. These tests were also performed with a significance level of *p* < 0.05.

## 3. Results and Discussion

### 3.1. Film Analysis

#### 3.1.1. Thickness, Moisture Content, and Water Solubility

Thickness is an important criterion to use when selecting the appropriate kinds of packaging and directly affects the water solubility and mechanical properties of packaging films. Based on Table 1, all the films achieved the desired thickness. The thickness of the NaCas film was 55% and 17% higher than that of CS and CS/NaCas films, respectively. Similar findings were reported by Kristo et al. [45], who reported that NaCas edible films displayed higher thickness compared with pullulan, blend, and bilayer films with pullulan edible films. The thickness of the edible film can be affected by the unique nature of colloidal compounds and the interaction between the constituents of the edible film [46]. According to Japanese industry standards, a desired edible film thickness must be less than 0.25 mm [47].

The moisture contents of pure CS, pure NaCas, and CS/NaCas films are stated in Table 1. The highest moisture content (24.8 ± 1.2%) was displayed by the CS film compared with the NaCas film and CS/NaCas film (Table 1). Similar results were obtained by Jiang et al. [48], who also reported that chitosan–sodium dodecyl sulfate composite films had a lower moisture content compared with pure chitosan films. This was attributed to the binding of both polymers, resulting in the limited availability of free water. Ideal packaging should maintain low moisture content ranging from 16 to 24% to allow for protecting the packaged product during its shelf life [49]. Hence, the CS/NaCas composite film in this study possessed the desired moisture content value that has the least effect on food safety and quality.

Water solubility is another important parameter used to determine the potential application of biopolymer film as moist food packaging and is usually influenced by the chemical structure of the film [50]. The water solubilities of pure CS, NaCas, and CS/NaCas composite films are stated in Table 1. The pure NaCas film had the highest water solubility (55.47 ± 1.94%) among the edible films in this study. An edible film with high water solubility is equivalent to one with low water resistance and poor stability in water [51]. This is due to its random coil structure that allows for more interaction between NaCas polymers and the water. 

In comparison to NaCas films, chitosan films had a lower solubility in water due to the strong interaction between polymer chains that limit the interaction with water [52]. Nevertheless, films obtained by blending CS and NaCas had the lowest water solubility (27.6 ± 3.9%), indicating that composite films would exert better water resistance as food packaging. This observation was consistent with Córdoba and Sobral [53], who reported that the addition of sodium caseinate to gelatin/chitosan blend films resulted in films with a reduced water solubility (29.5 ± 0.6%). There might be strong interactions between NaCas and chitosan when forming polyelectrolyte complex films, leading to fewer sites where the water molecules can bind in the polymer matrix [14].

In addition, the pH value of the film-forming suspension is also important when it comes to controlling the compatibility and solubility of the polymers. According to Kurek et al. [13], the blending of chitosan with whey protein yields a complex and heterogeneous structure during the film drying process. This was attributed to the rapid evaporation rate of acetic acid solution during the film-drying process, which led to a dramatic change in the pH of the film-forming suspension that induced incompatibility, explaining the final heterogeneous microstructure observed through scanning electron microscopy. Therefore, in this study, the interactions between CS and NaCas might have yielded a complex orientation in the polymer chains, thereby lowering the water solubility to a desirable value.

#### 3.1.2. Opacity and Color

The transparency of edible films is an important trait regarding the sensorial aspect, allowing for an increase in the overall acceptance by consumers, as it mimics the polymeric packaging materials that are transparent [54]. Table 2 shows the opacity values of pure chitosan, pure sodium caseinate, and composite edible films. The highest opacity was demonstrated by the composite film (7.40 ± 0.65). This could have been due to the interaction between CS and NaCas that limited the light that was able to pass through, causing the transparency of the film to decline. In agreement with this finding, Kurek et al. [13] stated that the lower pH from acetic acid could cause the incompatibility between CS and NaCas, leading to limited binding sites, and thus increasing its opacity. This is further supported by Azaza et al. [55], who reported that the opacity of chitosan films increased with the addition of *Sardinella aurita* protein isolate (SrPI).

Generally, a film with low opacity is preferable because the high transparency allows the customer to view the content inside the packaging. However, a high opacity film may be required to pack light-sensitive foods, such as meats, dairy products, and nuts, to prevent nutrient loss via photooxidation and extend the product’s shelf life. The optical value of CS/NaCas composite film in this study was comparable to CS/polyvinyl alcohol/fish gelatin edible film with an opacity value of 7.16 (A600/mm) [56]. Nevertheless, the high opacity of CS/NaCas composite films in this study could be suitable for the packaging of light-sensitive food products.

The color of an edible film could have an impact on the visual appearance of the food product upon application. Table 2 summarizes all color parameters, including the L*, a*, b*, and ∆E values of pure and composite CS/NaCas edible films. The blending of CS and NaCas increased the L* of the films, indicating that the CS/NaCas composite films were whiter. The difference in lightness between the pure and composite edible films could be due to the interactions between polysaccharides and proteins [57]. Indeed at acidic pH, electrostatic interactions between cationic chitosan and anionic sodium caseinate were already reported [15].

The composite film showed higher a* (redness–greenness) and b* (yellowness–blueness) values. This observation was supported by Wu et al. [12], who reported that the a* and b* parameters of films increased with the chitosan and carboxymethyl concentration increase in pullulan blended films. The increase in color intensity and yellowness of films may be attributed to the addition of chitosan in blended films providing more free amino groups, leading to a browning resulting from the Maillard reaction [12,14]. Moreover, the composite edible films showed the highest total color difference among the edible films (Table 2). Similarly, Lyu et al. [58] also reported that polycaprolactone–grapefruit seed extract composite edible films had a high total color difference compared with a pure polycaprolactone edible film. 

#### 3.1.3. Mechanical Analysis

Mechanical properties are essential for the adequate design of biopolymer-based packaging films that must have a certain degree of resistance. The tensile strength (TS), the elongation at break (EAB), and Young’s modulus are the key indicators to consider when testing the ability of edible films to be used as packaging [59]. The TS is the maximum load that the film can handle before it breaks, while the EAB value of an edible film represents its capability to resist shape changes without crack formation. Young’s modulus is the fundamental measure of the film stiffness or rigidity of the material [60].

Table 3 shows the TS, the EAB, and Young’s modulus of the CS, NaCas, and CS/NaCas edible films. The pure CS and NaCas films exhibited a higher TS than the CS/NaCas composite films. The lower TS of the composite films could have been due to the low pH of the acetic acid, causing NaCas to aggregate and form matrices that are less compact [13]. Nevertheless, the TS value of composite films obtained in this study met the minimum Japan Industry Standard, which is 3.92 MPa [61]. Several factors could also affect the TS of an edible film, which include the concentrations of plasticizer, biopolymer, and additives that can alter and interfere with the intermolecular reaction [62].

In this study, the CS/NaCas composite films exhibited 9.1% and 92.1% higher EAB values than pure CS and NaCas films, indicating that the composite film was more elastic. This was likely due to the blending of CS and NaCas forming bonds, thereby reducing the movement between polymers, hence increasing the flexibility of the films. The EAB of the CS/NaCas edible films was considered very good according to the Japan Industrial Standard, where 10–50% is considered good, while more than 50% is considered very good [63]. Similar results were reported by Volpe et al. [14], who made films with a CS blend with NaCas in a 1:1 (*w*/*w*) ratio that showed a higher EAB than pure chitosan films (i.e., 37 ± 11% vs. 62 ± 3%), leading to more flexible films.

Furthermore, the composite films had the lowest Young’s modulus compared with pure NaCas and CS edible films in this study. This shows that the composite film had the lowest TS, the highest EAB, and the lowest Young’s modulus. A similar trend was reported by Lin et al. [5], who observed that NaCas edible films with the lowest Young’s modulus also had the highest EAB and the lowest TS. This shows that the low Young’s modulus in the composite edible film reflected its flexibility.

### 3.2. Optimization of the Edible Film Characteristics

In general, edible films had a thickness of less than 0.3 mm. Hence, the thickness of the CS/NaCas composite film was considered adequate [64]. The low moisture content and water solubility of the composite films could limit the growth of foodborne pathogens and prevent the disintegration of the edible films when in contact with high humidity or food products with high water activity [65]. Although the high opacity of CS/NaCas composite films limits the view of the food product, it offers protection for light-sensitive foods [66]. Furthermore, the high flexibility characteristics of a CS/NaCas composite film also allow the biodegradable packaging to form the desired shape according to the food products and to be more resistant to breaking [21].

### 3.3. Addition of Lb. fermentum into the Edible Film

#### 3.3.1. Physical Properties

Table 4 displays the thickness, moisture content, and water solubility of the edible films with and without *Lb. fermentum*. The addition of probiotic cells did not significantly (*p* > 0.05) change the thickness of the CS/NaCas composite films (Table 4). Similarly, Soukoulis et al. [16] did not observe changes in the thickness of gelatin-based films blended with different soluble prebiotic fibers after the addition of *Lb. rhamnosus* GG cells. According to Nisar et al. [67], the addition of probiotics in edible films has no significant impact on the thickness due to the insignificant volume changes in the film-forming suspensions. 

The moisture content of an edible film can affect the survivability of probiotic cells in the edible film [68]. Table 4 shows that the presence of probiotic cells did not affect (*p* > 0.05) the moisture content of the composite films. This is consistent with Ebrahimi et al.’s [69] observation that the addition of *Lactococcus lactis* had no significant effect on the film moisture. Similar moisture contents ranging from 11.9 to 19.2% were measured in both probiotic-incorporated and control alginate-pectin edible films [51].

Furthermore, the incorporation of probiotics had no significant effect (*p* > 0.05) on the film’s solubility (Table 4). This finding is in agreement with Kanmani and Lim [46], who also reported that no significant difference was found between the water solubility of probiotic-incorporated and control pullulan/starch-based edible films due to their similar chemical composition. The lower water solubility of probiotic edible films reported in the present study (< 31%) in comparison to Pereira et al. [21] and Shahrampour et al. [51] (> 70%) indicated its higher water resistance as food packaging.

The opacity and color of the composite edible films with or without *Lb. fermentum* are presented in Table 5. Based on Table 5, the addition of probiotics did not result in color changes in the composite films. A similar observation was also reported by Soukoulis et al. [16] and Shahrampour et al. [51] upon the addition of *Lb. rhamnosus* and *Lb. plantarum* to edible films, respectively. As the difference in the ΔE value between a composite film with and without probiotics was not significant (*p* > 0.05), human eyes cannot detect the color difference between both films [70]. The film opacity was also not affected by the incorporation of probiotics (Table 5). This result is aligned with Namratha et al. [71], who reported that the entrapment of probiotic *Enterococcus faecium* Rp1 did not significantly increase the opacity of casein-based edible films. While the transparency of the edible films was affected by the type and combination of the polymers, as shown in Figure 1, the transparency of composite films with or without *Lb. fermentum* was considered visually indistinguishable (Figure 1C,D).

#### 3.3.2. Mechanical Properties

The effects on the TS, the EAB, and Young’s modulus upon the addition of probiotics in the CS/NaCas composite film formulation are stated in Table 6. The film’s TS decreased significantly (*p* < 0.05) (Table 6). Similar results were reported by Pereira et al. [72], where the incorporation of probiotics into a whey protein isolate edible film also resulted in a reduction in the TS. This was most probably due to the presence of probiotic cells interrupting the cohesiveness of polymer chains, hence less force was needed to break the film. However, the addition of probiotics did not markedly change the EAB value of the composite films in this study, which was similar to the observations of Pereira et al. [72]. Furthermore, the TS of CS/NaCas–*Lb. fermentum* also met the minimum Japan Industry Standard, which is 3.92 MPa [61].

A significant reduction in Young’s modulus was observed when the probiotics were added to the CS/NaCas composite film formulation (*p* < 0.05), indicating more flexibility than the composite films without the probiotic. A similar trend was also observed by García-Argueta et al. [73], where the addition of lactic acid bacteria interacted with the gelatin and inulin and significantly affected the Young’s modulus values of the films (*p* < 0.05). It is hypothesized that the probiotics could interact with the polymer, which would result in a lower Young’s modulus and films with higher elasticity [39].

### 3.4. Survivability of Probiotics after Film Drying

To maximize the health benefits offered by the probiotics, the viable cell count had to achieve at least 6 log_10_ CFU/g upon consumption. Figure 2 illustrates the impact of the drying and production process on the viability of *Lb. fermentum*. This study showed that the viable counts of *Lb. fermentum* dropped after film drying from 8.96 log_10_ CFU/g to 6.65 log_10_ CFU/g after incorporation into the CS/NaCas film. This result was aligned with Ma et al. [74], where the *Lactococcus lactis* culturable population in films was reduced by 1 to 2 log_10_ CFU/g after the film-drying process. The reduction in probiotic cells after being incorporated into edible films might be due to the stress that results from the drying and mixing processes [75]. The mixing process of *Lb. fermentum* in the film-forming process may induce mechanical stress on the cell membrane [76]. On the other hand, bacterial cells are comprised of 70–95% of intracellular water. Hence, the removal of water from cells during the air-drying process affects the fluid osmotic pressure and might cause cell injury [75]. 

Soukoulis et al. [16] reported that the incorporation of protein into the starch edible film can significantly reduce the loss of culturable *Lb. rhamnosus* GG population from 1.71 log_10_ CFU/g to 0.91–1.07 log_10_ CFU/g after evaporation–dehydration. The higher loss found in this study could have been due to the absence of a protective agent during the drying process [77]. Nevertheless, it was observed that the composite film’s probiotic *Lb. fermentum* population after drying still exceeded the minimal value for a probiotic, which is 6 log_10_ CFU/mL according to Pereira et al. [21]. Therefore, the composite films can still be considered good carriers for *Lb. fermentum*.

### 3.5. Antibacterial Test

The antibacterial activity of the *Lb. fermentum* composite film was determined against two foodborne pathogenic bacteria, which were *Escherichia coli* O157:H7 (Gram-negative) and *Staphylococcus aureus* (Gram-positive). The zone of inhibition around the composite films with and without *Lb. fermentum* against *E. coli* and *S. aureus* are shown in Table 7. It was observed that the films without probiotics inhibited the growth of *E. coli* and *S. aureus*. The antibacterial properties of chitosan-based composite films were already reported by Pereda et al. [78]: both chitosan–gelatin composite films and film-forming suspensions showed inhibitory effects against both *E. coli* and *Listeria monocytogenes*. The antibacterial properties of the composite films could be correlated to the presence of chitosan. The charges present along the chitosan chains were reported to interact with the ionic groups of bacteria and provoke the hydrolysis of the peptidoglycans in the microorganisms’ cell walls [79].

Upon the addition of *Lb. fermentum*, the zone of inhibition around chitosan–sodium caseinate composite films was 44.8% and 41.3% higher compared with the zone of inhibition around the same edible films without probiotics for *E. coli* and *S. aureus*, respectively. The antibacterial properties of *Lactobacillus*-containing sodium alginate, sodium carboxymethylcellulose, and collagen-based films against *S. aureus* population were already reported [74]. The antibacterial properties of the probiotic-containing films were correlated to the production of bacteriocin by probiotic cells and the ability of the edible film to retain the active metabolites [20]. Similar findings were reported by Heredia-Castro et al. [80], where the bacteriocin produced by *Lb. fermentum* was able to exert bacteriostatic effects against both Gram-positive (*S. aureus*) and Gram-negative pathogens (*E. coli*). The study found that the bacteriocin had peptides with high hydrophobic residues and a net negative charge. It was hypothesized that the high hydrophobicity may have enhanced the membrane disturbance in pathogens, while the negative charge density was responsible for the increased electrostatic bond between the peptides and the cell membrane. 

To exert antibacterial properties, a sufficient amount of probiotics is required in order to release bacteriostatic metabolites and inhibit the growth of pathogens [81]. Furthermore, probiotic cells could also inhibit the growth of pathogens by competing for available carbohydrates in the medium. By utilizing the available carbohydrates, probiotic cells could also create a low-pH environment by releasing lactic acid, thereby inhibiting pathogen growth [82,83]. Past studies reported a final viability of 6–9 log_10_ CFU/mL for probiotics after their incorporation into film-forming suspensions [27,46,72]. In addition, Hashemi and Jafarpour [22] reported that Konjac-based edible films incorporated with 6.4–7.1 log_10_ CFU/mL *Lb. plantarum* displayed antimicrobial properties against yeasts and molds. In the present study, the > 6.6 log_10_ CFU/g *Lb. fermentum* in the composite chitosan–sodium caseinate film was able to exert antibacterial properties against both *E. coli* and *S. aureus*.

## 4. Conclusions

In this study, the blending of NaCas and CS was able to produce a composite film with better flexibility and a good water barrier compared with pure NaCas or CS films. The addition of probiotics did not significantly affect the physical or mechanical properties of the CS/NaCas composite films, except for the tensile strength and Young’s modulus. The *Lb. fermentum*–CS/NaCas composite films had further improved flexibility, and the *Lb. fermentum* viable population exceeded 6.6 log_10_ CFU/g and exerted interesting antibacterial properties against *E. coli* and *S. aureus*. This shows the potential application of CS/NaCas composite films as a sustainable food packaging to minimize the excessive use of traditional synthetic plastic packaging. Moreover, the antibacterial properties of the probiotic composite film may further improve the shelf life and safety of the food products upon application. Future studies could explore different types of probiotics as the active ingredient, as well as the storage stability of composite films and their applications on food products as food packaging.

## Figures and Tables

**Figure 1 foods-11-03583-f001:**
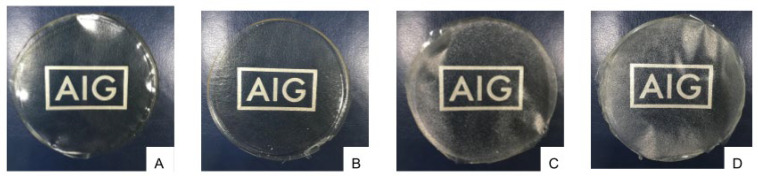
Visual authentication of the transparency for pure chitosan (**A**), sodium caseinate (**B**), chitosan/sodium caseinate composite film (**C**), and chitosan/sodium caseinate composite film with the addition of *Lb. fermentum* (6.65 log_10_ CFU/g) (**D**).

**Figure 2 foods-11-03583-f002:**
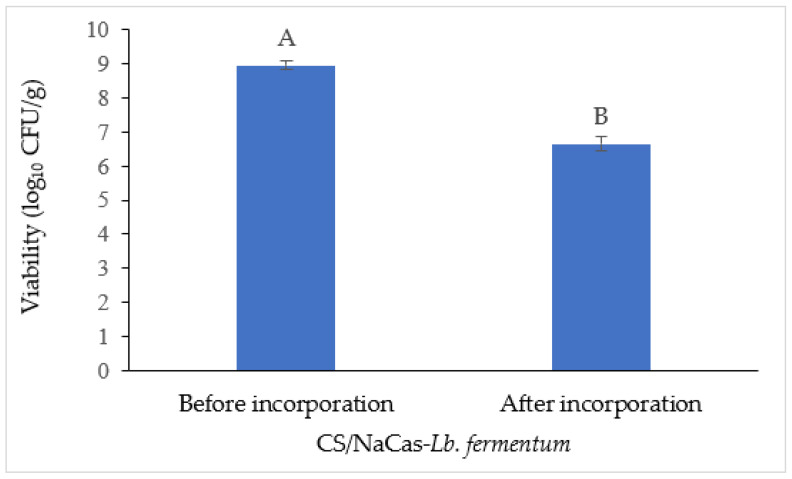
Viability of *Lb. fermentum* after incorporation into chitosan (CS)/sodium caseinate (NaCas) composite edible films. The results are expressed as mean ± standard deviation (*n* = 3). Different letters (**A**,**B**) mean that there were significant differences via a paired *t*-test.

**Table 1 foods-11-03583-t001:** Thickness, moisture content, and water solubility of chitosan, sodium caseinate, and composite edible films.

Edible Films	Thickness (mm)	Moisture Content (%)	Water Solubility (%)
CS	0.09$\;\pm \;0.01{\;}^{C}$	24.8 ± 1.2 ^A^	32.9 ± 1.9 ^B^
NaCas	0.14$\;\pm \;0.00{\;}^{A}$	10.0 ± 0.8 ^C^	55.5 ± 1.9 ^A^
CS/NaCas	0.12 ± 0.01 ^B^	17.2 ± 0.4 ^B^	27.6 ± 3.9 ^B^

The results are expressed as mean ± standard deviation (*n* = 3). Different letters (^A–C^) in the same column indicate a significant difference (*p* < 0.05) via ANOVA and Tukey’s post hoc test. CS, chitosan; NaCas, sodium caseinate; CS/NaCas, composite of chitosan and sodium caseinate.

**Table 2 foods-11-03583-t002:** Opacity and color parameters of chitosan and sodium caseinate edible films.

Edible Films	Opacity (A/mm)	L*	a*	b*	ΔE
CS	2.22 ± 0.01 ^B^	8.53 ± 1.02 ^C^	−0.65 ± 0.06 ^A^	−0.97 ± 0.08 ^B^	8.62 ± 1.01 ^C^
NaCas	1.65 ± 0.16 ^B^	13.41 ± 0.95 ^B^	−0.90 ± 0.12 ^B^	−1.85 ± 0.24 ^C^	13.58 ± 0.90 ^B^
CS/NaCas	7.40 ± 0.65 ^A^	21.20 ± 1.12 ^A^	−0.50 ± 0.07 ^A^	−0.42 ± 0.06 ^A^	21.25 ± 1.14 ^A^

The results are expressed as mean ± standard deviation (*n* = 3). Different letters (^A–C^) in the same column indicate a significant difference (*p* < 0.05) via ANOVA and Tukey’s post hoc test. CS, chitosan; NaCas, sodium caseinate; CS/NaCas, composite of chitosan and sodium caseinate films.

**Table 3 foods-11-03583-t003:** Tensile strength, elongation at break, and Young’s modulus of chitosan and sodium caseinate edible films.

Edible Films	Tensile Strength (MPa)	Elongation at Break (%)	Young’s Modulus (MPa)
CS	41.5 ± 3.0 ^B^	96.9 ± 3.1 ^A^	0.43 ± 0.04 ^B^
NaCas	57.8 ± 5.1 ^A^	8.4 ± 0.4 ^B^	6.92 ± 0.66 ^A^
CS/NaCas	28.5 ± 2.5 ^C^	106.6 ± 7.5 ^A^	0.27 ± 0.01 ^B^

The results are expressed as mean ± standard deviation (*n* = 3). Different letters (^A–C^) in the same column indicate a significant difference (*p* < 0.05) via ANOVA and Tukey’s post hoc tests. CS, chitosan; NaCas, sodium caseinate; CS/NaCas, composite of chitosan and sodium caseinate films.

**Table 4 foods-11-03583-t004:** Thickness, moisture content, and water solubility of composite edible films with or without *Lb. fermentum*.

Edible Films	Thickness (mm)	Moisture Content (%)	Water Solubility (%)
CS/NaCas	0.12 ± 0.01 ^A^	17.23 ± 0.41 ^A^	27.59 ± 3.86 ^A^
CS/NaCas–*Lb. fermentum*	0.11 ± 0.01 ^A^	17.94 ± 2.18 ^A^	30.80 ± 3.02 ^A^

The results are expressed as mean ± standard deviation (*n* = 3). Different letters (^A^) in the same column indicate a significant difference (*p* < 0.05) via an independent *t*-test. CS/NaCas, composite of chitosan and sodium caseinate films.

**Table 5 foods-11-03583-t005:** Opacity and color parameters of the composite edible film with or without *Lb. fermentum*.

Edible Films	Opacity (A/mm)	L*	a*	b*	ΔE
CS/NaCas	7.4 ± 0.6 ^A^	21.2 ± 1.1 ^A^	−0.50 ± 0.07 ^A^	−0.42 ± 0.06 ^A^	21.2 ± 1.1 ^A^
CS/NaCas–*Lb. fermentum*	8.7 ± 0.9 ^A^	21.7 ± 0.9 ^A^	−0.37 ± 0.04 ^A^	−0.31 ± 0.05 ^A^	21.7 ± 0.9 ^A^

The results are expressed as mean ± standard deviation (*n* = 3). Different letters (^A^) in the same column indicate a significant difference (*p* < 0.05) via an independent *t*-test. CS/NaCas, composite of chitosan and sodium caseinate films.

**Table 6 foods-11-03583-t006:** Tensile strength, elongation at break, and Young’s modulus of composite edible films with or without *Lb. fermentum*.

Edible Films	Tensile Strength (MPa)	Elongation at Break (%)	Young’s Modulus (MPa)
CS/NaCas	28.46 ± 2.45 ^A^	106.57 ± 7.51 ^A^	0.27 ± 0.01 ^A^
CS/NaCas–*Lb. fermentum*	20.94 ± 2.48 ^B^	88.80 ± 9.12 ^A^	0.24 ± 0.01 ^B^

The results are presented as mean ± standard deviation (*n* = 3). Different letters (^A,B^) in the same column indicate a significant difference (*p* < 0.05) via an independent *t*-test. CS/NaCas, composite of chitosan and sodium caseinate.

**Table 7 foods-11-03583-t007:** Antibacterial activity of composite edible films with or without *Lb. fermentum*.

Edible Films	*Escherichia coli*	*Staphylococcus aureus*
CS/NaCas	0.37 ± 0.06 ^Bb^	0.47 ± 0.06 ^Bb^
CS/NaCas–*Lb. fermentum*	0.67 ± 0.06 ^Aa^	0.80 ± 0.05 ^Aa^

The results are presented expressed as mean ± standard deviation (*n* = 3). Different uppercase letters (^A,B^) in the same column indicate that the means were significantly different (*p* < 0.05) via an independent *t*-test. Different lowercase alphabet letters (^a,b^) in the same row indicate that the mean values were significantly different (*p* < 0.05) via an independent *t*-test. CS/NaCas, composite of chitosan and sodium caseinate films.

## Data Availability

The data presented in this study are available on request from the corresponding author.

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
