# Peer review of "Chitosan–Sodium Caseinate Composite Edible Film Incorporated with Probiotic Limosilactobacillus fermentum: Physical Properties, Viability, and Antibacterial Properties"

_foods, 2022, doi:10.3390/foods11223583_

Round 1

Reviewer 1 Report

COMMENTS TO THE AUTHORS:

This is an interesting manuscript describing Chitosan – sodium caseinate composite edible film incorporated with probiotic Lactobacillus fermentum. But the following points need to be done by the authors:

1.        Line 15 and throughout the text: Lactobacillus fermentum, I now known as Limosilactobacillus fermentum. A tool can be used to quickly search for new names of bacteria: (http://lactobacillus.ualberta.ca/).

2.        Introduction: please include some related studies, including the benefits of edible films incorporating probiotics.

3.        Line 89: follow the journal guideline for citation.

4.        Line 116: A good, precise protocol for centrifugation instructs you to use the g force rather than RPMs because the rotor size might differ, and g force will be different while the revolutions per minute stay the same.

5.        Line 133: Once the complete name of a bacteria has been written out once, the genus name can be abbreviated to just the capital letter. Please Follow this rule and correct in the whole text.

6.        M&M section: line 134: why you added L. fermentum suspension at 8.96 log10 CFU/mL into the film?

7.        Please use uniform units CFU/mL or CFU mL-1.

8.        M&M section: Antibacterial Activity, line 196: why the bacterial cells adjusted to 105-106 CFU/mL? Please clarify.

9.        Line 426: Survivability of Probiotics after Film Drying: Please improve the discussion section and the reason for the decrease in probiotic cells.

10.    Line 449: Antibacterial Test: Since the cell walls of gram-positive and gram-negative bacteria are different, what is the exact reason that the composite film has the same effect on both bacteria?

11.    Include a possible mode of action of probiotic strain against pathogenic microorganisms.

12.    Make sure references are in the form of Journal requirements.

13.    Some correction of English (e.g. plural / singular + verbs) and formatting (spaces) is suggested. Moreover, in several parts of the manuscript serial commas (i.e. Oxford commas) are missing. Please review the manuscript carefully.

Reviewer 2 Report

The study entitled “Chitosan-sodium caseinate composite edible film incoporated with probiotic L. fermentum: Physical properties, viability, and antinbacterial properties had been well-written with novelty. However, the article needs to be improved according to following comments. 

- The bacteria L. fermentum was reported to have antimicrobial activity against various microorganisms; gram-negative or gram positive bacteria. The explanation about the mechanisms of the bacteria to have an antimicrobial effects remained still unclear. Please add short explanation related to this issues. 

- Line 102-103, The author mentioned lactic acid production, which might reduced pH to inhibit the bacteria, but the bacteria grown in the film, not in the food. Is there any potential migration of bacteria from film to food matrix to inhibit the bacteria growth? 

- Line 116-118, In the preparation of Culture. The supernatant contained the probiotic cells of the bacteria with final cell count of 10^8 to 10^9 CFU/mL. Was the bacteria added directly to film formulation or the bacteria stored first? If the bacteria was stored before usage in film formulation, how Authors control the population of isolated bacteria from the bacterial death? 

- Line 135 to 136, the films with bacterial suspension were dried at 40 oC  fro 48 h. How to ensure the bacteria remain active during drying? Is there any potential in reduction of the bacterial population?

- Is there any measurement to indicate the bacteria grown homogeneously in the film surface? Since the homogeneity of the bacterial population is important to ensure all sides of the film have good potential to inhibit the bacteria. 

- Eq 1 and 2 are definitely similar. Please have a look at and change the right one

- Line 232-233, what is the relation of nutritional component and microbiological damage to obtained result? It is better to remove the sentence

- The sample used to be added the bacteria was CS/NaCas. However, the tensile strength was low, which might reduce the functionality of edible film in package the food, so the film can be easily damaged during storage and distribution of packaged food. This point needs to be clarified and considered for its application. 

- According to the mechanical properties, visual authentication for CS/NaCas-lb. fermentum, the film was significantly lower than the film without lb. ferementum. In terms of the antibacterial properties, the film with lb. fermentum is better to be applied, but in terms of the protection, I do not think the film can be effective to protect perishable foods. How do authors consider this circumstances for the effectiveness of the film in protecting perishable foods?

- Survivability of Probiotics has been investigated, where the film drying reduce the bacteria viable counts by around 2 log10 CFU/g. Is there any potential of bacterial growth in the film in hours or days after the drying process? 

It is suggested for author to clarify and modify the articles according the above comments. The article has a good novelty, but it needs a major revision.

Round 2

Reviewer 1 Report

The authors have generated the suggested changes with clear and concrete responses to each of the comments I made. As a result, the manuscript is much improved. I have no objection to publication in Foods.

Reviewer 2 Report

The comments have been adressed well by Authors.